# Modelling and Control of Mechatronics Lines Served by Complex Autonomous Systems

**DOI:** 10.3390/s19153266

**Published:** 2019-07-24

**Authors:** Florin Dragomir, Eugenia Mincă, Otilia Elena Dragomir, Adrian Filipescu

**Affiliations:** 1Department of Automation, Computer Science and Electrical Engineering, Valahia University of Târgoviște, 130024 Târgoviște, Romania; 2Department of Automation and Electrical Engineering, Dunărea de Jos University of Galați, 800008 Galați, Romania

**Keywords:** complex autonomous mobile robots, collaborative systems, assembly/disassembly mechatronics lines, Petri nets

## Abstract

The aim of this paper is to reverse an assembly line, to be able to perform disassembly, using two complex autonomous systems (CASs). The disassembly is functioning only in case of quality default identified in the final product. The CASs are wheeled mobile robots (WMRs) equipped with robotic manipulators (RMs), working in parallel or collaboratively. The reversible assembly/disassembly mechatronics line (A/DML) assisted by CASs has a specific typology and is modelled by specialized hybrid instruments belonging to the Petri nets class, precisely synchronized hybrid Petri nets (SHPN). The need of this type of models is justified by the necessity of collaboration between the A/DML and CASs, both having characteristics and physical constraints that should be considered and to make all systems compatible. Firstly, the paper proposes the planning and scheduling of tasks necessary in modelling stage as well as in real time control. Secondly, two different approaches are proposed, related to CASs collaboration: a parallel approach with two CASs have simultaneous actions: one is equipped with robotic manipulator, used for manipulation, and the other is used for transporting. This approach is correlated with industrial A/D manufacturing lines where have to transport and handle weights in a wide range of variation. The other is a collaborative approach, A/DML is served by two CASs used for manipulation and transporting, both having simultaneous movements, following their own trajectories. One will assist the disassembly in even, while the other in odd workstations. The added value of this second approach consists in the optimization of a complete disassembly cycle. Thirdly, it is proposed in the paper the real time control of mechatronics line served by CASs working in parallel, based on the SHPN model. The novelty of the control procedure consists in the use of the synchronization signals, in absence of the visual servoing systems, for a precise positioning of the CASs serving the reversible mechatronics line.

## 1. Introduction

Lately, the industry is faced with new global evolution, driven by the technological progress. This improvement extends to all industrial domains and triggers the evolution of new generations of advanced flexible production systems and new methods of centralized management distributed or supervised. In addition, this involves the evolution of new types of robots and processing machine tools and the need for efficient transport and manipulation systems [1]. In this context, the possible approaches are related to the concepts of assignment, planning and execution of tasks on assembly/disassembly (A/D) [2,3], manufacturing lines served by mobile platforms equipped with manipulators, with emphasis on the planning of operations.

Assembly mechatronics lines are flow-oriented production system where the productive units perform operations on workstations, which may be configured as serial, parallel, circular, U-shaped, cellular or two-sided lines. The pieces pass through the stations successively, help by a conveyor belt [1] while the disassembly operations involve selection of the reusable parts from the disassembled products [4]. The A/D processes are real-time, complex control systems, involving multiple operation, conditions, and tasks. A/D plans are made up of parts or subassemblies that are fitted together. Some research topics include A/D representations, work-cell planning, sequence planning, etc. Off-line task planning is a large area encompassing a diverse set of planning methodologies capable of producing a detailed operation plan, including planning sensory action, planning manipulator action, planning the trajectory of mobile robots [5], rough motion planning, fine motion planning and other planning [6].

In the last decade the industry has faced a new global evolution, driven by the technological progress, involving also the evolution towards new types of robotic systems, novel processing machine tools and efficient transport and manipulation systems [7,8,9,10]. Most of the studies are based on the increase of the number of manufacturing operations on the same equipment and the increase of productivity. Both of them have an important impact on the quality of the final product. On the other hand, it is well known that the quality of the product and the manufacturing process are tightly bounded [11,12].

Flexibility and process optimization have attracted the attention of researchers in the field. The robotic serving systems are designed to efficiently and simultaneously serve multiple processes (shared resources) or to adapt the handling and transport capabilities to the dimensional characteristics and weight of the handled parts. Types of collaborative robotic systems work by combining two service functions: moving function—mobile robotics, and disassembled component manipulation function. The researchers in the field pay a special interest to process optimization and ensuring its flexibility. In this respect, service robotic systems are designed to serve efficient and simultaneously several processes (collaborative systems), or to adapt their possibilities of handling and transport to the characteristics of the manipulated pieces. The existing topologies of robotic systems with collaborative actions rely on the optimization of two function: one is related to the robotic platform equipped with a powerful system for handling movement and the other one refers to handling of the pieces. In the paper approach, for optimizing the complete disassembly cycle time, the two complex autonomous systems (CASs) will work collaboratively. Both CASs and wheeled mobile robots (WMRs) equipped with robotic manipulators (RMs), work simultaneously on assigned posts, even or odd for each of them. Through collaborative control strategy, CAS collision is avoided. The two CASs are moving in a coordinated and non-competitive fashion, with permanent respect for the relative position: one behind the other. Due to their simultaneous action, the full disassembly cycle time will be minimized.

Assembly/disassembly mechatronics line (A/DML) are hybrid systems served by WMRs. These have hybrid characteristics, consisting of continuous or discrete behaviors. Frequently, hybrid Petri nets (HPNs) are the tools proposed to model this type of systems. Modelling of reversible A/DMLs serviced by WMRs equipped with RMs and the generalized modelling of these systems have been proposed in a series of papers previously published [13,14,15,16]. In this framework, the article proposes an extended approach for the modelling of A/DML from the perspective of service control typologies for CASs: autonomous CASs, CASs with collaborative or parallel actions. The work reported here focuses on the hybrid aspect of the processes, in modelling approach, and take in consideration work scenarios consider the working scenarios for CASs servicing. The hybrid control system of an A/DML serviced in parallel actions by two CASs, is able to control in real-time control the entire process, according to the proposed strategy.

The rest of the paper is organized as follows: in Section 2, the framework of the proposed work is presented: the architecture and technical characteristics of automated control systems for Hera&Horstmann A/DML, as well as the CASs integrated in A/DML: Pioneer3-DX and PatrolBot. The synchronized hybrid Petri nets (SHPN) models for the proposed A/DML served by CASs, based on scheduling of disassembly tasks in parallel approach, and the offline simulation are proposed in Section 3. Section 4 includes scheduling of disassembly tasks in collaborative approach, the SHPN models and the offline simulation of Hera& Horstmann A/DML assisted by two collaborative CASs. The real-time control of A/DML served by two parallel CASs is detailed in Section 5. The Conclusion section summarizes the added values of the paper, as well as the work in progress research directions.

## 2. Framework: Hera & Horstmann A/DML Served by CASs

The extended approach for modelling and control of mechatronics lines served by collaborative CASs uses a previous configuration of A/DML mechatronics line [5], Hera & Horstmann served by CASs working in parallel or in collaborative relations Figure 1a.

The flexible line includes five individual workstations with different tasks: carrying and transporting, pneumatic workstations, conveyor belt, sorting unit, test station and warehouse. The work part carrier is used for carrying and transporting the four-piece work part on a conveyor belt system. The work part carrier is equipped with six-bit identification, which provides a large number of possible codes, read out by inductive sensors. The four-piece work part enables workflow operations such as assemblies, testing, sorting, storage, and disassembling.

The processing stations S1, S2, S3, S4, S5 Figure 1a have warehouse parts associated, in each of them being located one of the components. The last station (S6) is the final product storage place. Each station is equipped with position sensors for a precise localization to the corresponding warehouse. The workpiece parts positioning on the conveyor belt is made by actuators equipped with pneumatic pistons controlled by a pneumatic system. The components to be assembled are Figure 1b: work part carrier (base platform) (1), body (2), cover (3), metal cylinder (4) and plastic cylinder (5).

The architecture of the automation system is a distributed one and consists of SIEMENS Simatic S7-300 with a series CP 314C–2 DP processor and a communication module SIEMENS CP 343–2. The automation system is connected to the PROFIBUS DP which interfaces with auxiliary modules I/O type SIEMENS ET200S IM 151–1 stations distributed on each of the flexible system for A/D. Each SIEMENS ET200S-IM 151–1 module has digital and analogue I/O signals taking signals from transducers and giving commands to actuators. A SIEMENS Simatic HMI TP 177 operator panel is connected on the PROFIBUS DP terminal, through which the system status can be checked and an execution process of assembly or disassembly can be implemented.

The CAS’s integrated into A/DML are: Pioneer3-DX (two driving wheels and one rear free wheel, 2DW/1FW) and PatrolBot (two driving and two rear free wheels, 2DW/2FW), each of them equipped with odometric system Figure 1a. In addition, an on-board embedded microcontroller is able to read position information and send it, via a WI-FI link, to a remote PC according to a specific protocol. The remote PC computes control input and sends it to CAS. Also, the remote PC sends the data to the assembly line PLC [17]. The CAS is equipped with a RM, Pioneer 5-DOF Arm, with three articulations and one gripper paddle.

## 3. SHPN Modelling and Offline Simulation of A/DML Served by Two Parallel CASs

The need for the SHPN model is justified by the necessity of collaboration between the mechatronics line and the CASs that serve it. Precisely in this approach, the hybrid Petri net (HPN), obtained from synchronized HPN without synchronized signals from sensors, is modelled, simulated and tested in autonomous mode. The compatibility is needed because the mechatronics line and CASs have characteristics and physical constraints that should be considered.

The proposed HPN model is indispensable for simulation and represents the preceding stage of real-time control implementation. As a result of the simulation of the HPN model, it is possible to monitor the evolution of the integrated system, A/DML served by CASs, in the state space, as a result of the transients. Therefore, the evolution is consistent with the constructive elements. The inputs of HPN, imposed in modelling stage are: the scheduling of the operations on A/DML, the durations of those ones, the distances and the CASs movement durations, the manipulation durations for each operation, the estimated precise positioning times of the manipulator for taking the piece from the disassembly location and storing it in the corresponding warehouse.

The precise positioning times represent a major uncertainty in our approach because of the existing constructive constraints that could compromise the real-time control. The existing solutions for this problem are based on eye to hand or eye in hand servoing systems. The implementation of this type of control represent for us a target in the nearest future. Until then, we propose for real time control implementation, the HPN model improved with synchronization signals, able to trigger the transitions of manipulator for its precise positioning for take-off the piece from the disassembly station and storing it in the warehouse.

### 3.1. Scheduling of Disassembly Tasks in Parallel Approach

In this approach, two CASs with parallel actions are specific to A/D processes of large pieces requiring appropriate handling systems. The control of A/DML served by CASs is based on the hierarchical model of reversible A/DML proposed in [2,18,19,20,21,22].

The model is an oriented graph described with the Petri Nets (PN) formalism, in accordance with the discrete events system (DES) concepts. The SHPN model associated with A/DML served by one CAS and/or two CASs, integrate different typologies of PN models, specific to A/D, transport and handling Figure 2.

The modelling process take into consideration the manufacturing line architecture presented in Figure 3 and the following assumptions:
$W_{j}$ is j warehouse and $D_{j}$ is j workstation for “J” stage of disassemblyThe two general CASs serve simultaneously each disassembly workstationCAS1 executes in each j workstation only handling operationsCAS2 waits for each j workstation to be loaded or downloaded by CAS1CAS1 and CAS2 travel simultaneously in parallel directions with the A/DML.

Generally, the A/DML dynamics served by CAS is described using TPN (temporal Petri nets) tools and typologies, if are considered only assembling operations. For disassembly operations, these are replaced by SHPN models interfaced with TPN models witch describe better the disassembly operations correlated with CASs actions, through synchronized signals.

The SHPN associated with the A/DML process served by two CASs with parallel action is interfaced with real processes by external synchronization signals, used to validate the transitions/actions to which are assigned. $E_{dd(j)}^{1}$ and $E_{dd(j)}^{2}$ are signals from the sensors or control structures, used for synchronization between A/DML and CASs [23,24]. The SHPN model is design in the following assumption: when the uncertainties appear caused by errors or defaults in functioning of A/DML, the mechatronics line stops to be solved the problems issued. The system is not reset in this case and will continue the functioning from the stop point.

The collaboration of CAS1 and CAS2 with parallel actions Figure 4 is ensured by the following synchronization:
⚬$E_{\mathrm{dd}(j)}^{1}$ → STOP line (End of Disassembling $D_{j}$), *START CAS 1*⚬$E_{dd(j)}^{2}$ → CAS1_ PICKING UP disassembled component and Closure of gripper, START Conveyor belt (START Line)⚬$E_{dd(j)}^{0}$ → End of CAS1′s manipulation operations. START parallel travel of CAS1, CAS2.

### 3.2. Process’s Modelling in Parallel Approach

The disassembly process is split into elementary tasks (each one corresponding to a single disassembly operation served by two CASs, as proposes in [2]. These tasks are model using TPN nets and are associated to a disassembly operation served by CASs with parallel actions, mainly used for transporting and manipulation actions. These are represented using SHPN models. Each model is reinitialized when CASs pass from “J” to “J+1” task Figure 5.

In this case, the SHPN model corresponding to elementary cycles “j” are shown in Figure 5. SHPN model for disassembly sequences served by two mobile robots with parallel actions, in our approach is based on an algorithm in which the movement actions of the two CASs are perfectly synchronized and parallel. The CAS2 robot is designed to carry the CAS1 loaded with components, between workstations. CAS1 performs only manipulation actions of disassembled components: “loading” the CAS2 with each disassembled component, or “unloading” and depositing it in the warehouse. When CAS1 travel between workstations, it is “empty”, because it is robot dedicated to manipulation actions.

The models contain state variables associated with the items evolution: disassembly part, CAS1 and CAS2. After the sequence: CAS1_Piece manipulation, piece dropping from $D_{j}$ to CAS2, the control variable P16 will synchronize the sequentially of the actions: “${\mathrm{Travel}\;D}_{j}\to W_{j}$” for both mobile robots. The same approach is proposed for “CAS1_Piece manipulation, Piece dropping from CAS2 to $W_{j}$”, followed by the change of the state variable P19 and the synchronization of the CASs actions: “${\mathrm{Travel}\;W}_{j}\to D_{j}+1$”.

### 3.3. Simulation of the Process

For stage “J”, a detailed SHPN model is proposed, integrating all the operations made CAS1 and CAS2 in a disassembly task. Since CAS2 executes only travel actions between workstations, followed by stationary actions, in order to simplify this approach, the SHPN detailed model for stage “J” considered only the actions of CAS1. In this context, Figure 6a presents a simplified representation of the A/DML reversible mechatronics line, serviced by CAS1.

The following notations are used: S1,…,S6 are the workstations; the distances between disassembly locations: $D_{L}^{1},\ldots {,D}_{L}^{5}$; the Cylinders/Covers/Bodies/Bases WAREHOUSE correlated with the positioning points of CAS1 Figure 6b within the manipulation/transport process: ${\mathrm{Rd}}_{1},\ldots {,\mathrm{Rd}}_{8}$.

CAS1 carries the component from the place where disassembly occurs to the appropriate storage warehouse. A complete disassembly cycle Figure 6b contains the following tasks: disassembling, recovered component and storage in the related warehouses. For a CAS with parallel actions, the tasks to be complete are:
-Stage 1: CAS1′s elementary cycle for travel/storage “cylinder 1” in the Cylinders Warehouse-Stage 2: CAS1′s elementary cycle for travel/storage “cylinder 2” in the Cylinders Warehouse-Stage 3: CAS1′s elementary cycle for travel/storage “cover” in the Covers Warehouse-Stage 4: CAS1′s elementary cycle for travel/storage “body” in the Bodies Warehouse-Stage 5: CAS1′s elementary cycle of reposition in S1-Stage 6: CAS1′s elementary cycle of reposition in STOP

In Figure 6b are indicated, for each stage of the disassembly process, the trajectories of mobile robots (points of piece pick-up, travel trajectory and points of piece dropping) as well as the remaining distances to go by mobile robots until they stop. In offline simulation, due to the complexity of the model, in order to have accurate and detailed results, the disassembly process was split sub-processes representing successive stages of the disassembly (Figure 7, Table 1).

These values presented in Table 1 are validated by simulation for Stage 2 Figure 7 and Stage 4 Figure 8. Since the model associated with *assembly process* has a simple TPN typology, it was supposed that the simulation would not highlight outstanding results regarding viability, boundedness, reversibility and eventually bottlenecks or model stability. For each stage of disassembly process, the elementary models with hybrid typology has been separated from the general model: SHPN corresponding to an elementary “J” disassembling stage served by CAS.

The hybrid characteristic of the approach is given by the discrete and continuous variations of CAS states, and the states of the disassembled parts. The SHPN model for the j-th stage was customized for stage 2 Figure 7 and stage 5 Figure 8, where:
Continuously variable states of CAS1: Pc_CAS1_(r),…,CAS1_(r+3) are associated with the continue changes in the CAS1 status, respectively the variable “remaining distance” of CAS1 in relation to the final moment of complete disassembly Table 1.The CAS’s dynamics in discrete approach are indicated through the discrete variables associated to the places: Pd_CAS1_(s),…,Pd_CAS1_(s+8) which corresponds to the manipulation operations (pick-up and dropping).Pdd(k),…,Pdd(k+5) are variable states associated with “J” stage of piece disassembling.

The simulation results of SHPN models associated of each elementary “J” disassembling stage have the initial and final markings related distances proposed in Table 1: the initial mark contains the variable “Travelled distance of CAS until the end of disassembly”—$M_{0}\left(\mathrm{Pc}\_\mathrm{CAS}\_(r)\right)$, and the final mark contains the variable “remaining distance of CAS until the end of disassembly”—$M_{0}(\mathrm{Pc}\_\mathrm{CAS}\_(r+3))$. For the elementary “J+1” disassembling stage, the initial marking is $M_{0}\left(\mathrm{Pc}\_\mathrm{CAS}\_(r)\right)=M_{0}\left(\mathrm{Pc}\_\mathrm{CAS}\_(r+3)\right)$ corresponding to the previous “J” stage Table 1.

In order to integrate timed transitions, synchronization signals were added in the SHPN model, according CAS1 timeframes: handling (pick-up and dropping), standby, and travel. The travel has a general route: the disassembly post, storage warehouse, next station of disassembly. The states evolution in the global model of SHPN result from the evolution sequences in the six basic models, correspond to the six stages. All simulations highlight bounded models, viable and accessible, in terms of marking and without any bottlenecks. Thus, the global model SHPN has the same properties and in addition, the reversibility characteristic.

Figure 9 illustrates the CAS1′s dynamics for stage 2, its continuous displacement (the markings of Pc_CAS1_(r)=Pc_CAS1_(r+3)) the absence of delays or bottlenecks. For each moment in time, these variables represent the distance to be travelled by the CAS1, in relation to the STOP point of complete disassembly cycle.

The temporal sequence of each CAS’s manipulation actions, corresponding to the marking of Pd_CAS1_(s), Pd_CAS1_(s+2), Pd_CAS1_(s+6), Pd_CAS1_(s+7) is monitored Figure 10.

## 4. SHPN Modelling and Offline Simulation of A/DML Served by Two Collaborative CASs

In this approach the A/DML process is served by two CASs who works in collaborative mode. This structure is appropriate the optimization of a complete disassembly cycle.

### 4.1. Scheduling of Disassembly Tasks in Collaborative Approach

In this approach, the following assumptions were made Figure 11
$W_{j}$ is j warehouse and $D_{j}$ is j workstation for “J” stage of disassembly;The two resources-CASs serve successively the disassembly stations;CAS1 and CAS2 move in parallel directions with the A/DML;CAS1 serves the odd stations with $\left\{D_{2k+1}\right\},\;k=0,\ldots ,n$;CAS2 always works behind CAS1 (synchronization signals coordinate their action and avoid collisions);CAS2 serves the even stations with $\left\{D_{2k}\right\},\;k=0,\ldots ,n$;CAS2 is positioned for each moment by START disassembly in $\left\{{D\;\prime }_{2k}\right\},\;k=0,\ldots ,n$;CAS2 executes for each disassembly operation the following travel cycle: ${D^{\prime }}_{2k}\to D_{2k}$
$D_{2k}\to W_{2k}$, $W_{2k}\to {D^{\prime }}_{2k+1}$, $D_{2k+1}\to {D^{\prime }}_{2k+1}$.

For both CASs, the duration of the actions and their relationship were done by the $\left\{E_{\mathrm{dd}(j)}^{1},\ldots {,E}_{\mathrm{dd}(j)}^{4}\right\}$ synchronization signals Figure 12. Coherence of collaborative actions is ensured by synchronization signals as follows:
$E_{\mathrm{dd}(j)}^{1}$ → End of Disassembling $D_{j}$, START CAS 1$E_{\mathrm{dd}(j)}^{2}$ → CAS1_Closure of gripper, START Conveyor belt (START Line)$E_{\mathrm{dd}(j)}^{3}$ → CAS1 in $D_{j+2}$, START CAS2$E_{\mathrm{dd}(j)}^{4}$ → *CAS2_Closure of gripper, START Conveyor belt (START Line).*

In the case of CASs with collaborative actions, the sequence of actions for j, j+1, j+2 were the following Figure 12
-Synchronization signal release $E_{\mathrm{dd}(j)}^{1}$: CAS1 component pick-up from $D_{j}$ and close gripper-Synchronization signal release $E_{\mathrm{dd}(j)}^{2}$: CAS1 travel on the route $D_{j}\to W_{j}$, CAS1 component dropping in $W_{j}$ and CAS1 travel on the route $D_{j}\to D_{j+1}$-Synchronization signal release $E_{\mathrm{dd}(j)}^{3}$: CAS2 repositioning from ${D^{\prime }}_{j+1}$ in $D_{j+1}$, CAS2 component pick-up from $D_{j+1}$ and close gripper-Synchronization signal release $E_{\mathrm{dd}(j)}^{4}$: CAS2 travel on the route $D_{j+1}\to W_{j+1}$, CAS2 component dropping in $W_{j+1}$, CAS2 repositioning from $W_{j+1}$ in ${W^{\prime }}_{j+1}$, CAS2 travel on the route ${W^{\prime }}_{j+1}$ → ${D^{\prime }}_{j+3}$, CAS1 continues to travel along the route $D_{j+1}\to D_{j+2}$.

CAS1 serves the stations with the even identification number $\left\{D_{2j}\right\},\;j=0,\ldots ,N$ while CAS2 served odd station $\left\{D_{2j+1}\right\},\;j=0,\ldots ,N$.

### 4.2. Process’s Modelling in Collaborative Approach

The collaborative aspect of our approach is based on the use of CAS1 who serves odd-numbered workstations, while CAS2 serves workstations with even identification number. In addition, for both CASs, the control algorithm integrates sequences dedicated to tasks synchronization control, in order to avoid collisions between them [5]. In the Figure 13, the SHPN model presents the algorithm of disassembly process served by two mobile robots for J, J+1, J+2 stages. ${\mathrm{Edd}}_{1}$, ${\mathrm{Edd}}_{2}$, ${\mathrm{Edd}}_{3}$ and ${\mathrm{Edd}}_{4}$ are synchronization signals used to control the synchronization of disassembly sequences with the actions of the two mobile robots with collaborative actions.

The proposed model, being a generalized one, can be tested for A/DML with N workstations, with the travel distances updating for CAS1 and CAS2. By simulation, it is possible to validate the control structure for whole system, A/DML served by two collaborative resources.

### 4.3. Offline Simulation of the Process

The SHPN models in collaborative approach, integrates the presented algorithms as well the synchronization signals for different activities were simulated in offline mode [5]. The results of simulation reported here, corresponding to the collaborative control showed in Figure 14. Here are illustrated the discrete states of the work piece (P20,…,P27), discrete states of CAS1 (P1,…,P7), discrete states of CAS2 (P8,…,P15) and the synchronization signals ${\mathrm{Edd}}_{1}$, ${\mathrm{Edd}}_{2}$, ${\mathrm{Edd}}_{3}$ and ${\mathrm{Edd}}_{4}$. For CAS1 (serves posts disassembly with even number $\left\{D_{2j}\right\},\;j=0,\ldots ,N$) and CAS2 (serves posts disassembly with odd number $\left\{D_{2j+1}\right\},\;j=0,\ldots ,N$), the representative states corresponding to the positions in the disassembly locations were selected $D_{j},\;j=\bar{1,N}$ or those of warehouses allocated $W_{i},\;i=\bar{1,N}$: ${\mathrm{CAS}1\_D}_{j}$, ${\mathrm{CAS}1\_W}_{j}$, ${\mathrm{CAS}1\_D}_{j+2}$, ${\mathrm{CAS}2\_D}_{j+1}$, ${\mathrm{CAS}2\_W}_{j+1}$, ${\mathrm{CAS}2\_D}_{j+3^{\prime }}$.

The results must be correlated with the planning actions of the three entities: CAS1, CAS2 and workpiece, in order to obtain the maximum timings for each activity.

Analyzing the simulation results Figure 14, it can be seen that for the states variation of CAS1, (P1,…,P7), the state variation CAS2 (P8,…,P15), as well as for the disassembled work piece (P20,…,P27) were successive and without delays, in accordance with the coordination of control by the synchronization signals.

## 5. Real-Time Control of A/DML Served by Two Parallel CASs

### 5.1. Real-Time Control of CASs Based on Kinematic Model

For controlling CASs, trajectory-tracking, sliding mode control (TT-SMC) is presented in [25]. The CASs: 2DW/1FW, Pioneer 3-DX and 2DW/2FW, Patrol Boot, presented in Figure 15, are controlled to follow a desired trajectory with an imposed velocity.

The control in trajectory tracking mode relies on the hypotheses that the real CAS follows the trajectory of a virtual CAS which moves with an imposed speed. In this assumption, the desired trajectory generated by virtual CAS Figure 16 is denoted with $q_{d}\left(t\right)={\left[\begin{array}{ccc} x_{d} & y_{d} & \theta_{d} \end{array}\right]}^{T}$.

The cinematic model of the virtual CAS (Figure 17) is:(1)$$\{\begin{array}{l} {\dot{x}}_{d}=v_{d}{\cos\theta }_{d}\\ {\dot{y}}_{d}=v_{d}{\sin\theta }_{d}\\ {\dot{\theta }}_{d}=\omega_{d} \end{array},$$
where $\left(x_{d},y_{d}\right)$ represent the Cartesian coordinates of geometric center, $v_{d}$ is the linear speed, $\theta_{d}$ is the orientation, and $\omega_{d}$ is angular speed. When CAS is controlled to follow the desired trajectory, on Ox and Oy axes appear tracking and orientation errors, as follows:(2)$$\left[\begin{array}{c} x_{e}\\ y_{e}\\ \theta_{e} \end{array}\right]=\left[\begin{array}{ccc} {\cos\theta }_{d} & {\sin\theta }_{d} & 0\\ -{\sin\theta }_{d} & {\cos\theta }_{d} & 0\\ 0 & 0 & 1 \end{array}\right]\;\left[\begin{array}{c} x_{r}-x_{d}\\ y_{r}-y_{d}\\ \theta_{r}-\theta_{d} \end{array}\right].$$

The following errors dynamics is given by the following equations:(3)$$\{\begin{array}{l} {\dot{x}}_{e}=-v_{d}+v_{r}{\cos\theta }_{e}+\omega_{d}y_{e}\\ {\dot{y}}_{e}=v_{r}{\sin\theta }_{e}-\omega_{d}x_{e}\\ {\dot{\theta }}_{e}=\omega_{r}-\omega_{d} \end{array}.$$

It is supposed that $\left|\theta_{e}\right|<\pi /2$, meaning that the CAS orientation isn’t perpendicular on desired trajectory. Considering the errors’s position, from (2), and their derivatives, from (3), the sliding surfaces are defined:(4)$$\{\begin{array}{c} s_{1}={\dot{x}}_{e}+k_{1}x_{e}\\ s_{2}={\dot{y}}_{e}+k_{2}y_{e}+k_{0}\mathrm{sgn}(y_{e})\theta_{e} \end{array}$$
where $k_{1}{,k}_{2}{,k}_{3}\geq 0$ are positives and constant parameters. The $x_{e}{,y}_{e}{,\theta }_{e}$ are errors from Equation (2). If $s_{1}$ converges to zero then $x_{e}$ converges to zero too. If $s_{2}$ converges to zero, then ${\dot{y}}_{e}=-k_{2}y_{e}+k_{0}sgn(y_{e})\;\theta_{e}$. If $y_{e}>0$ then ${\dot{y}}_{e}<0$ only if $k_{0}<k_{2}\left|y_{e}\right|/\left|\theta_{e}\right|$. It can be observed that if $y_{e}$ and ${\dot{y}}_{e}$ converge to zero then $\theta_{e}$ converges to zero. The surface derivatives
(5)$$\{\begin{array}{c} {\dot{s}}_{1}={\ddot{x}}_{e}+k_{1}{\dot{x}}_{e}\\ {\dot{s}}_{2}={\ddot{y}}_{e}+k_{2}{\dot{y}}_{e}+k_{0}sgn(y_{e}){\dot{\theta }}_{e} \end{array}$$
can be written in compact form as:(6)$$\dot{s}=-Q\;\mathrm{sgn}(s)-P\;s$$
where:(7)$$Q=\left[\begin{array}{cc} Q_{1} & 0\\ 0 & Q_{2} \end{array}\right],\;P=\left[\begin{array}{cc} P_{1} & 0\\ 0 & P_{2} \end{array}\right],\;Q\geq 0\;,\;P\geq 0,$$
(8)$$s={\left[\begin{array}{cc} s_{1} & s_{2} \end{array}\right]}^{T},$$
(9)$$\mathrm{sgn}(s)={\left[\begin{array}{cc} \mathrm{sgn}(s_{1}) & \mathrm{sgn}(s_{2} \end{array})\right]}^{T}.$$

From Equations (2)–(6) after some manipulation, the TT-SMC law is:(10)$${\dot{v}}_{c}=\frac{-Q_{1}\mathrm{sign}(s_{1})-P_{1}s_{1}-k_{1}{\dot{x}}_{e}-{\dot{\omega }}_{d}y_{e}-\omega_{d}{\dot{y}}_{e}+v_{r}{\dot{\theta }}_{e}\sin\theta_{e}+{\dot{v}}_{d}}{\cos\theta_{e}},$$
(11)$$\omega_{c}=\frac{-Q_{2}\mathrm{sign}(s_{2})-P_{2}s_{2}-k_{2}{\dot{y}}_{e}-{\dot{v}}_{r}\sin\phi_{e}+{\dot{\omega }}_{d}x_{e}+\omega_{d\cdot }{\dot{x}}_{e}}{v_{r}\cos\phi_{e}+k_{0}\mathrm{sgn}(y_{e})}+\omega_{d}.$$

Based on linear and angular speed of the CAS, can be computed right and left angular speed of the right and left wheel, respectively:(12)$$\left[\begin{array}{c} \upsilon_{d}\\ \upsilon_{s} \end{array}\right]=\left[\begin{array}{cc} \frac{1}{R} & \frac{L}{R}\\ \frac{1}{R} & -\frac{L}{R} \end{array}\right]\;\left[\begin{array}{c} v_{r}\\ \omega_{r} \end{array}\right],$$
where $x_{r}$ is CAS position along $O_{x}$; $y_{r}$ is CAS position on $O_{y}$; $\theta_{r}$ is the orientation of the CAS; $v_{r}$ is linear speed of the CAS; $\omega_{r}$ is the angular speed of the CAS. The speed of mobile platform is constant for every elementary cycle and depends on the load it is carrying. This represents the reference of sliding mode mobile platform control loop.

### 5.2. Control Structure in LabVIEW and Graphic User Interface (GUI)

The framework described in previous sections and the offline simulation results allow us to apply a control strategy. The A/DML reversible line served by two parallel CASs makes possible the operation, synchronization and real-time control of flexible manufacturing process, for a given production batch.

Assembly tasks synchronization, testing, decision support and disassembly in case of default issue, are controlled in real time mode, using LabVIEW software. The application design with this tool receives the monitored signals by sensors mounted along mechatronics line, trough data acquisition board (DAQ) and programmable logic controller (PLC) These signals are then used to start or stop the execution of certain tasks according to the planning and optimization goal. The hybrid modelling and model tests were described in the sections above, were need to correlate the dynamic discrete evolution of mechatronics line with continuous evolution of mobile platforms. These approach represents, among other, an added value point of this paper.

The communication between the programmable logic controller (PLC) integrated in the flexible mechatronics line and the workstation that synchronizes with the robotic platform is realized with data acquisition board DAQ NI USB-6008, device manufactured by National Instruments [25]. The digital outputs and inputs of the Siemens CPU 314C-2 DP programmable controller operate at 0–24 V voltage and the acquisition board works with voltages ranging between 0–5 V, a relay board. It is used to avoid damaging the plate with the associated links between the digital inputs/outputs of the PLC and the digital inputs/outputs of the acquisition board. The control application, build with LabVIEW software.

Figure 18 shows the block diagram of the application. It is composed of two local control loops, one for each entity: A/DML, respectively CAS equipped with RM.

### 5.3. Implementation and Tests Results

The execution of control algorithm generates “.txt” files containing the position of the mobile robot equipped with the manipulator, during the disassembly process. The application reads the variables from these files, updates the graphical interface with the new CAS position, and intervenes in the disassembly synchronization algorithm.

The communication between these two local loops is accomplished through a flexible line interface and a process calculator through a NI DAQ-6008 acquisition board [26] that collects and transmits data from the assembly and disassembly process and wireless communication on a TCP/IP protocol between the mobile robot and the processor. In this application the master program, implemented in LabVIEW, has the role of real time synchronizing of the two loops in the disassembly process. In our case, Hera & Horstmann A/DML is served by two CASs, one (CAS1) is equipped with RM, used for manipulating, and the other (CAS2) are used only for transport. Both robotic platforms serve A/DML in the disassembly process. The two robotic systems operate in synchronized parallel mode, as follows:-CAS1 equipped with RM takes over the component from the location where disassembly occurs and places it on the platform (CAS2)-CAS1 and CAS2 move simultaneously to the storage location, where RM takes over the component on the CAS2 platform and places it in the warehouse.

The application for monitoring the process is design in Visual C++ and runs on a remote laptop. It gives the possibility to monitors the A/D processes, and to execute and synchronize tasks on the three subsystems: A/DML, CAS1 equipped with RM and CAS2. The supervisor communicates with the two robotic systems through the TCP/IP protocol and with A/DML via a NI USB-6008 acquisition board, connected to the digital inputs/outputs of the PLC. It assigns: the tasks of the robotic system, in terms of movement between disassembly workstations and the warehouse storage; position the RM, controls gripper closure/opening; commands the conveyor stops and starts, and synchronizes A/DML actions with the CAS1, RM and CAS2.

Figure 19 shows data workflow in real-time control mode of Hera & Horstmann A/DML reversible line served by two parallel CASs.

The implementation of the models and application deployed and explained in previous section gives the possibility to evaluated the distances ($Rd_{1},\ldots ,Rd_{8}$) traveled by CAS1, CAS2 within a complete disassembly cycle of a product (Table 2 and Figure 6a).

Tests results for real time control of A/DML served by CASs with parallel actions, implemented using LabVIEW, are depicted in Figure 20.

Figure 20 shows the variables associated with continuous states of CAS1 (corresponding to variable position of CAS1) are correlated with discrete states of RM (gripper ON/OFF), in a complete disassembly cycle. Parallel and synchronized movements of CAS1 and CAS2, correlated with manipulation tasks (pick-up and dropping) of RM are shown in Figure 21.

The control algorithm of CASs in SM-TT was implemented in Visual Studio C++ and gives the possibility to identify the linear speed characteristics of CASs, for each stage of a complete disassembly cycle Figure 22.

The traveling speed of the CASs is 94 mm/s. Figure 19 shows the variations of CASs speed during a complete cycle. When the robots execute actions in parallel, there are time intervals when CAS1, Pioneer 5-DOF Arm, performs simulation of gripping/releasing the components of the finished product under disassembly. In these intervals the linear velocity of the CAS1 is zero. Figure 23 shows that the linear motion of the mobile robot attends negative values. This values correspond to the reverse stroke of the CASs when the cylinder is retracted and when the robot returns to its original position. A backward movement performs CASs when it returns to the original position immediately after simulating.

The trajectory imposed on the mobile robots (CAS1 and CAS2) is a linear one because it is desired to move in parallel, along the A/DML. The localization errors were calculated through offline localization simulations using collected sensor data. Figure 23 shows the history of the localization error for CAS1, variations in the sum of squares of positional errors, $x_{e}$ (longitudinal error) and $y_{e}$ (lateral error) while the robot is moving. The localization error is relatively large from about the 10th iteration to the 30th iteration and the localization error is also relatively large in those iterations. It is noted that the CAS1 follows the desired trajectory with small errors.

## 6. Conclusions

The added value of the paper is to reverse an assembly line, to be able to perform disassembly, using two CASs. In this goal, the characteristics and physical constraints of A/DML and of the two CASs serving it have been put into relation with the constraints of disassembly, transport and storage processes. In the same time, the physical characteristics of CASs have been correlated with the physical characteristics of the manipulated piece. These aspects lead the researchers towards two approaches describing the interactive working of CASs: parallel and collaborative modes.

In modelling approach, the interactive “parallel” actions of the CASs was introduced as solution for disassembly processes of heavy and large manipulated parts, while the collaborative interactive actions of the CASs was proposed to optimize the time for a complete disassembly cycle. The HPN modelling of A/DML served by two CASs has been defined based on the tasks planning and scheduling, proposed in the paper. The SHPN models obtained have demonstrated the need of tasks CASs’s synchronization with the sequential tasks on A/DML. This type of control strategy is related to industrial processes assisted by CASs, in absence of precise positioning visual servoing systems.

The control strategy was implemented for A/DML served by CASs with parallel actions. The supervised control proposed in the paper, have synchronised the A/DML with CASs’s manipulations and CASs’s movements in SM-TT control. It was designed in LabVIEW and the results of the real-time control tests have been presented in the paper.

The work in progress research directions are oriented towards the implementation of real-time control of A/DML served by CASs with collaborative actions, in a hybrid and hierarchical control architecture.

## Figures and Tables

**Figure 1 sensors-19-03266-f001:**
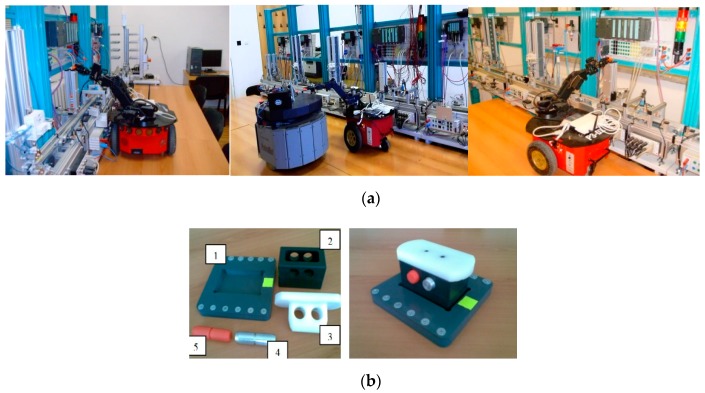
(**a**) Assembly/disassembly (A/D) stations and related components; (**b**) final product.

**Figure 2 sensors-19-03266-f002:**
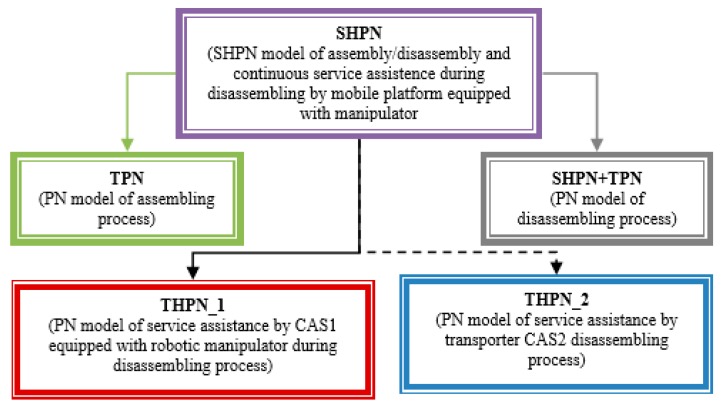
Synchronized hybrid Petri nets (SHPN) structure of A/DML CAS1 and CAS2.

**Figure 3 sensors-19-03266-f003:**
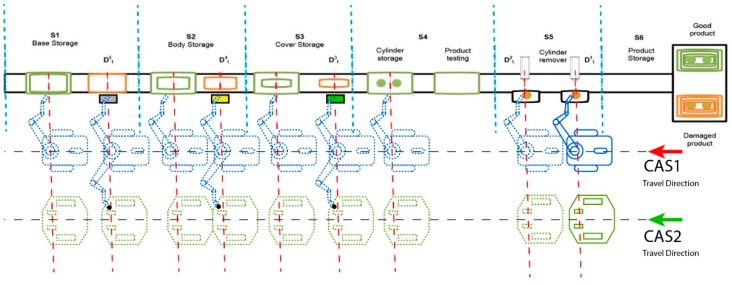
Hera &Horstmann A/DML served by two CASs with parallel actions.

**Figure 4 sensors-19-03266-f004:**
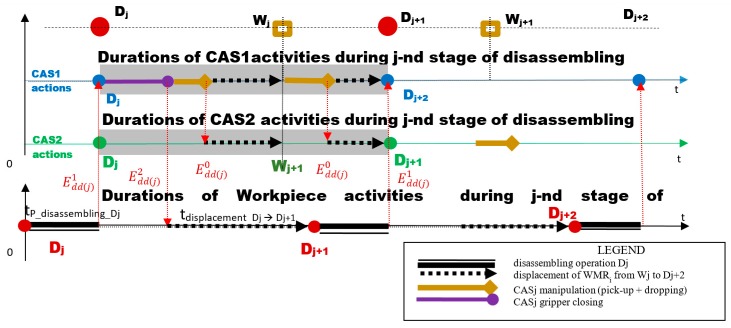
Sequence scheduling of elementary J disassembly stage served by CAS1 and CAS2 with parallel actions.

**Figure 5 sensors-19-03266-f005:**
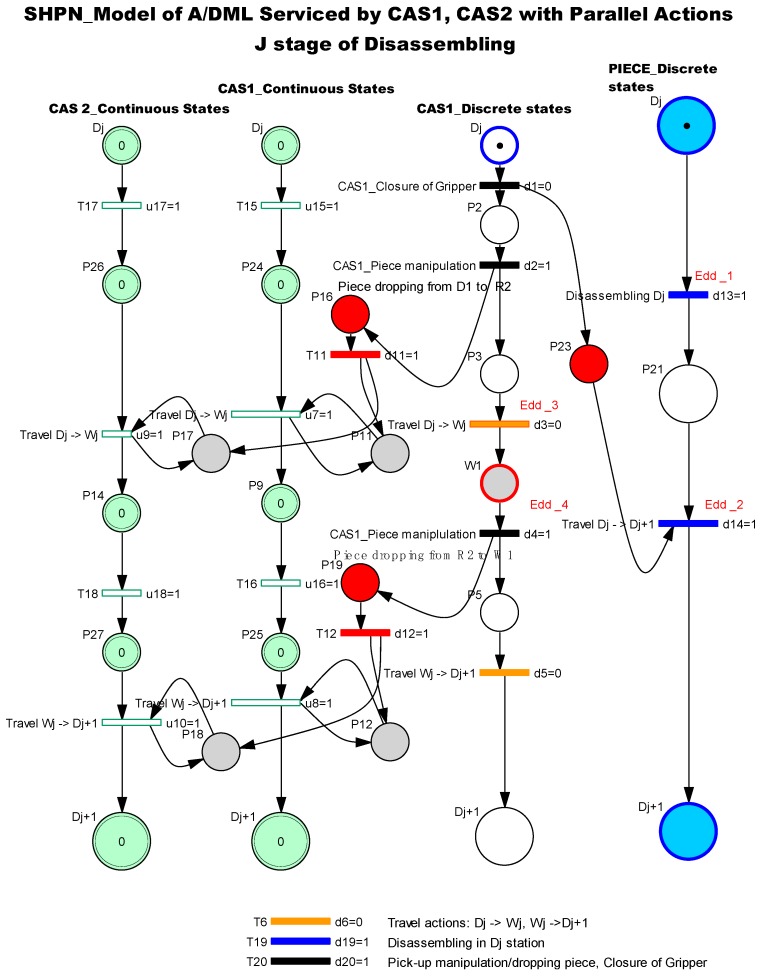
SHPN model for disassembly sequences served by two CASs with parallel actions.

**Figure 6 sensors-19-03266-f006:**
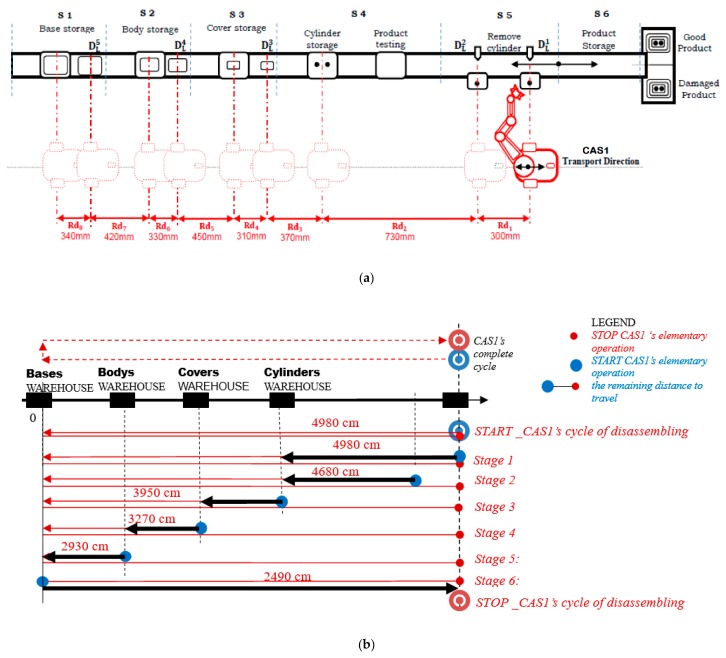
(**a**) Hera & Horstmann A/DML disassembly workstations and warehouses locations; (**b**) CAS1 complete disassembly cycle.

**Figure 7 sensors-19-03266-f007:**
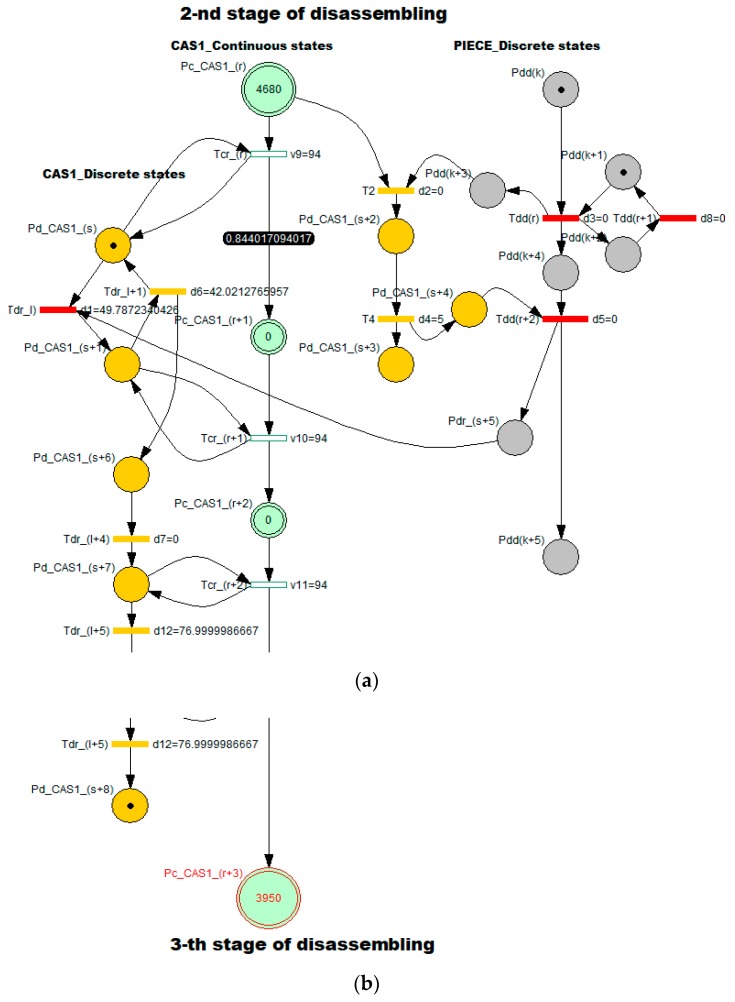
SHPN corresponding to stage 2 of “cylinder 2” disassembly: (**a**) initial marking; (**b**) final marking.

**Figure 8 sensors-19-03266-f008:**
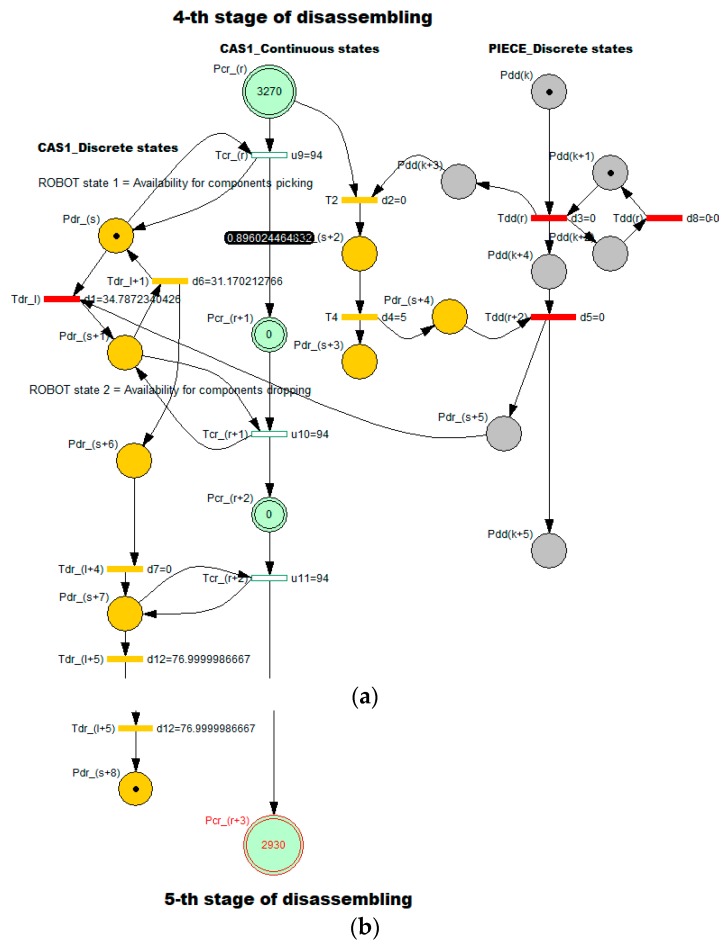
SHPN model corresponding to 4-th stage of “body” disassembly operation: (**a**) initial marking; (**b**) final marking.

**Figure 9 sensors-19-03266-f009:**
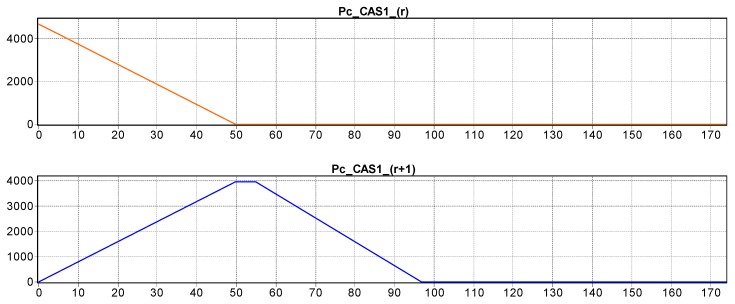
Continuous states evolution of CAS1/CAS2 travel positions corresponding to ‘body’ disassembly.

**Figure 10 sensors-19-03266-f010:**
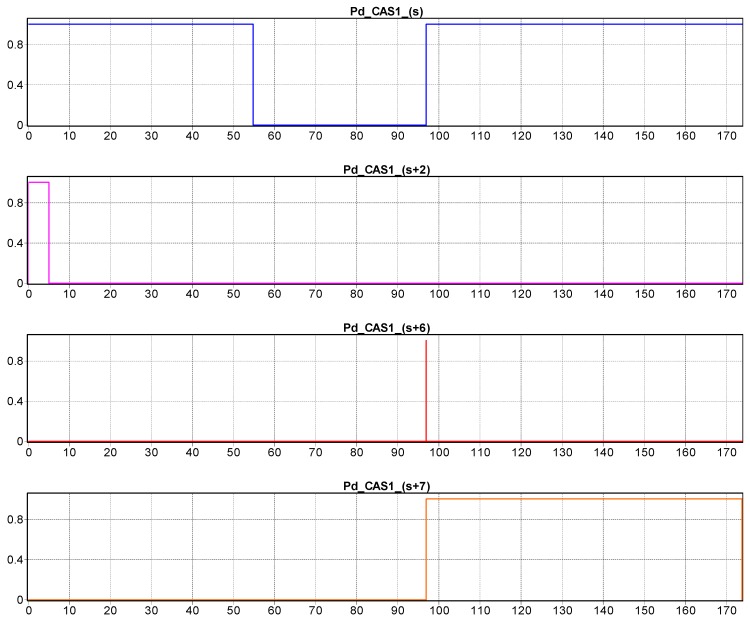
Discrete states evolution of CAS1, corresponding to “body” manipulations of disassembly operation.

**Figure 11 sensors-19-03266-f011:**
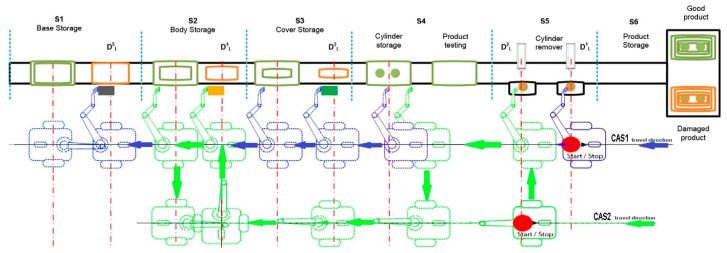
A/DML, Hera & Horstmann served by two CASs with collaborative actions.

**Figure 12 sensors-19-03266-f012:**
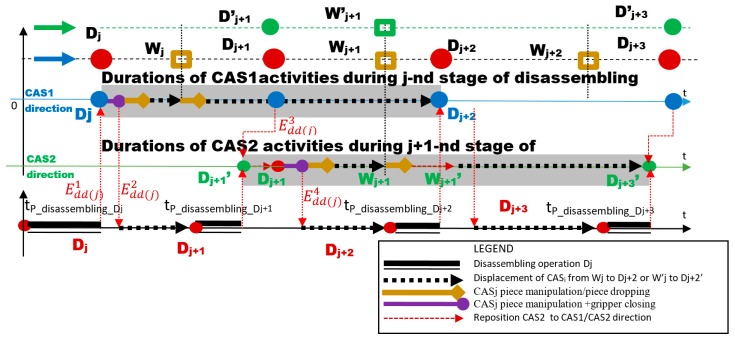
Sequence scheduling of elementary disassembly j, j+1 stages served by CAS1 and CAS2 with collaborative actions.

**Figure 13 sensors-19-03266-f013:**
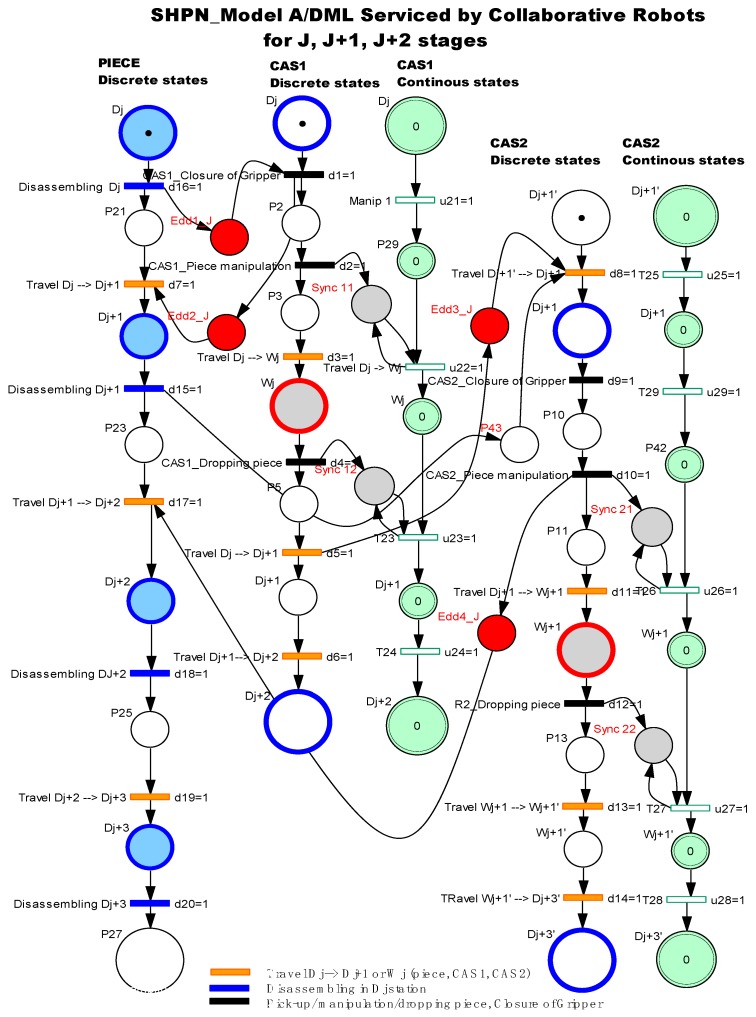
SHPN model for disassembly sequence served by two CAS1, CAS2 with collaborative actions.

**Figure 14 sensors-19-03266-f014:**
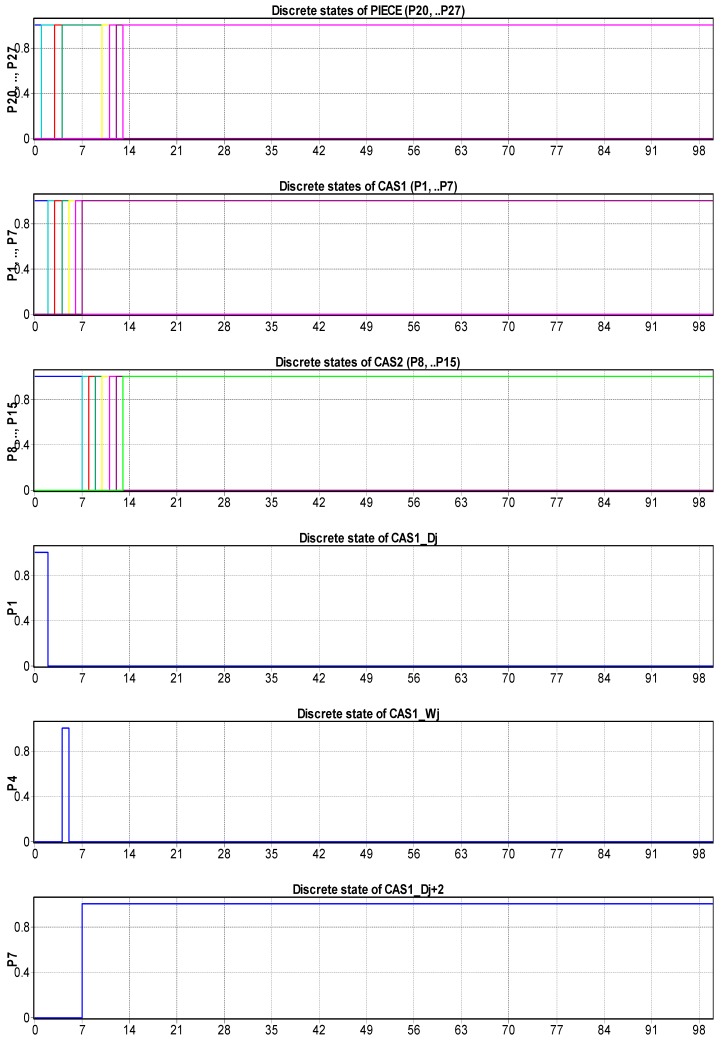
Discrete states of disassembled work piece, CAS1 and CAS2 in relation with synchronization signals for collaborative actions.

**Figure 15 sensors-19-03266-f015:**
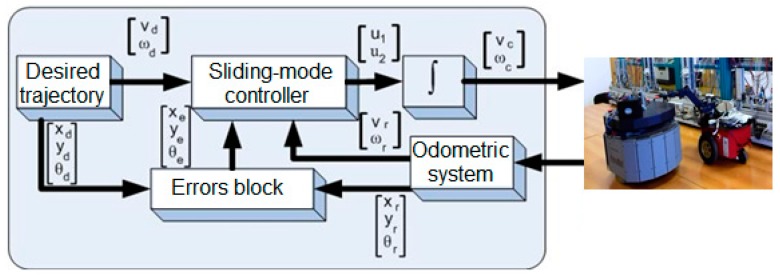
Closed loop trajectory-tracking, sliding mode control (TT-SMC) of the two CASs, 2DW/1FW, Pioneer 3-DX and 2DW/2FW, Patrol Boot.

**Figure 16 sensors-19-03266-f016:**
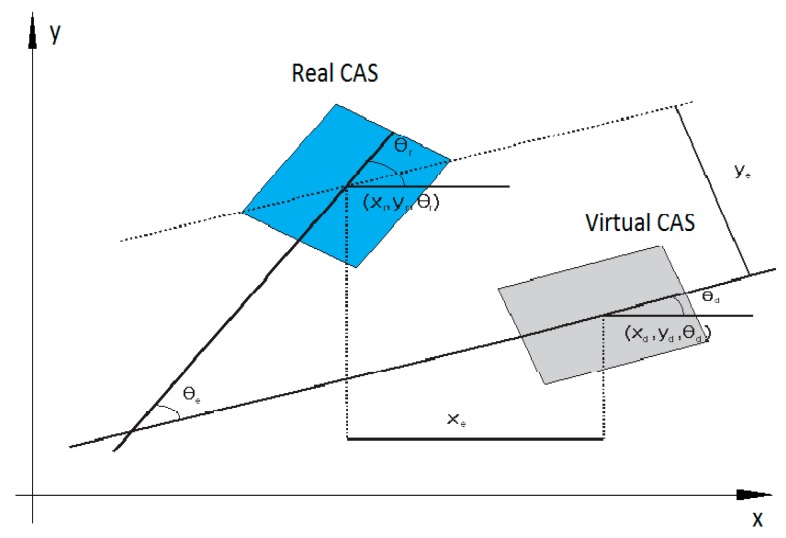
Tracking errors of the real and virtual CAS.

**Figure 17 sensors-19-03266-f017:**
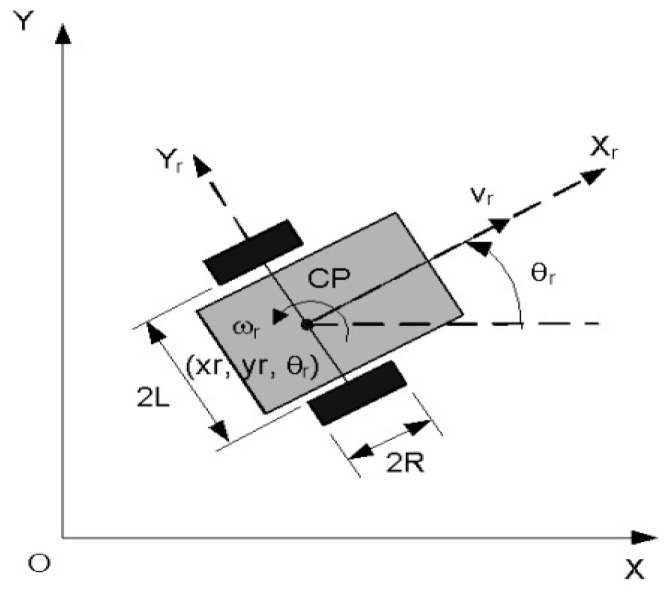
Cinematic model of CAS with 2DW/1FW or 2DW/2FW.

**Figure 18 sensors-19-03266-f018:**
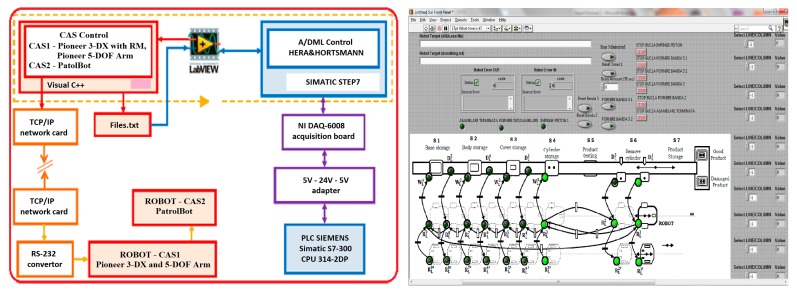
Control application of mechatronics line made with LabVIEW.

**Figure 19 sensors-19-03266-f019:**
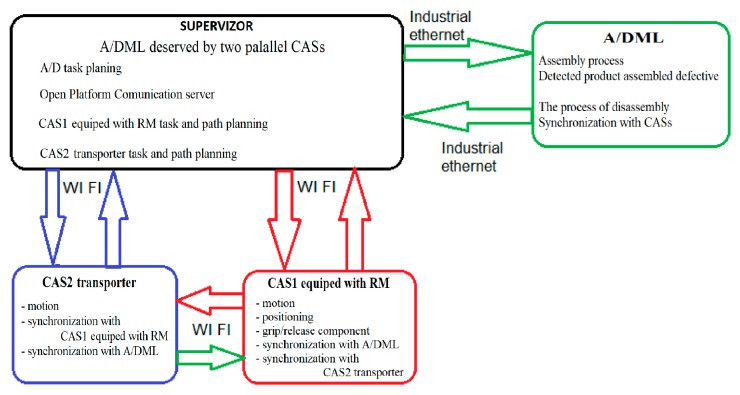
Data workflow in real-time control mode.

**Figure 20 sensors-19-03266-f020:**
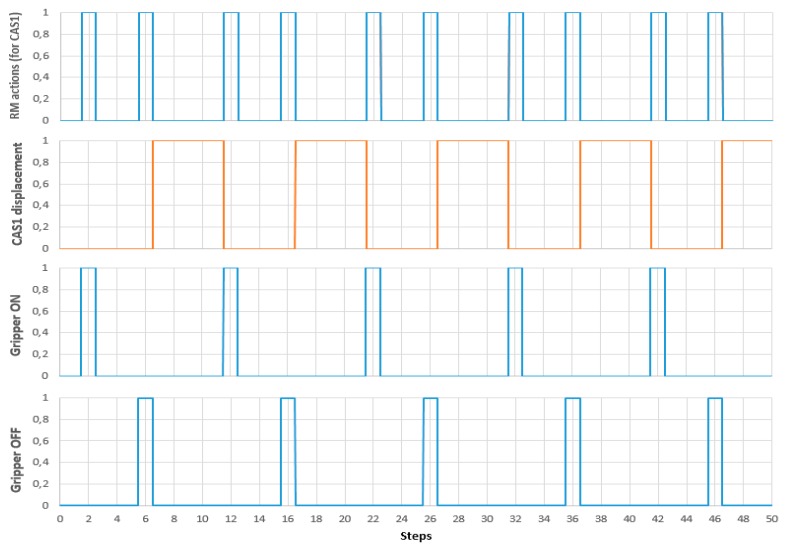
Control of CAS1 movement, correlated with manipulation tasks, in a complete disassembly cycle.

**Figure 21 sensors-19-03266-f021:**
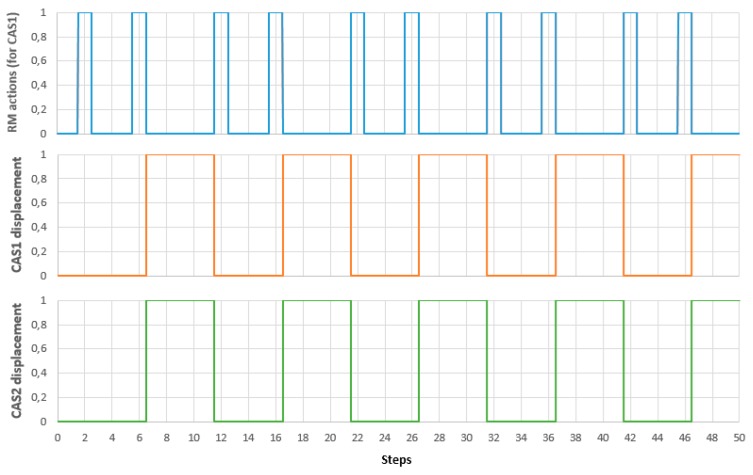
Control of CAS1 and CAS2 synchronized movement, correlated with manipulation tasks, in a complete disassembly cycle.

**Figure 22 sensors-19-03266-f022:**
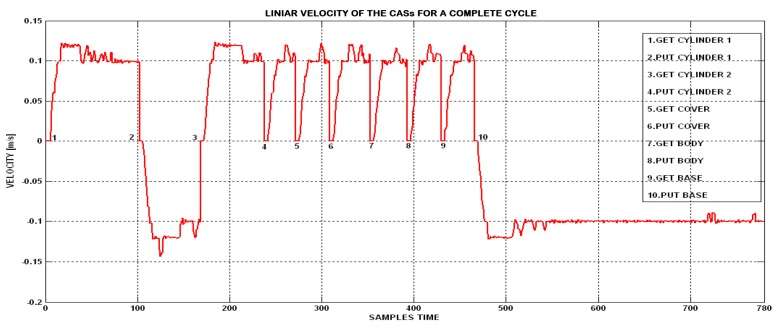
Linear velocity of CASs during a complete cycle of disassembly.

**Figure 23 sensors-19-03266-f023:**
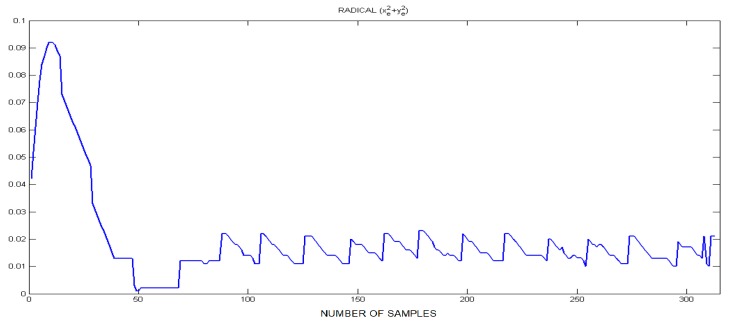
Position’s error—$x_{e}$ and $y_{e}$.

**Table 1 sensors-19-03266-t001:** “Remaining distance” for complex autonomous systems (CASs) travel according to the stages of a disassembly cycle.

The Stages of a Complete Disassembly Cycle		$M_{0}\left(Pc\_CAS\left(r\right)\right)\;\mathrm{Travelled}\;\mathrm{Distance}\;\mathrm{of}\;\mathrm{CASs}\;\mathrm{Until}\;\mathrm{the}\;\mathrm{End}\;\mathrm{of}\;\mathrm{Disassembly}$	$M_{0}\left(Pc\_CAS\left(r+3\right)\right)$ Remaining Distance of CASs
Stage 1	Disassembly “cylinder 1”	4980	4680
Stage 2	Disassembly “cylinder 2”	4680	3950
Stage 3	Disassembly “cover”	3950	3270
Stage 4	Disassembly “body”	3270	2930
Stage 5	Disassembly “palette”, CASs reposition in S1	2930	2490
Stage 6	CASs reposition in STOP point	2490	0

**Table 2 sensors-19-03266-t002:** “Distances travelled” by CASs for a disassembly cycle.

Disassembly Cycle	$Rd_{1}\;[\mathrm{mm}]$	$Rd_{2}\;[\mathrm{mm}]$	$Rd_{3}\;[\mathrm{mm}]$	$Rd_{4}\;[\mathrm{mm}]$	$Rd_{5}\;[\mathrm{mm}]$	$Rd_{6}\;[\mathrm{mm}]$	$Rd_{7}\;[\mathrm{mm}]$	$Rd_{8}\;[\mathrm{mm}]$	Total [mm]
Disassembly “cylinder 1”	300	730	730						1760
Disassembly “cylinder 2”		730	730						1100
Disassembly “cover”				310	450				760
Disassembly “body”						330	420		750
Disassembly “pallet”								340	340
back to the initial position	300	730	730	310	450	330	420	340	3250

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
