# Peer review of "Modelling and Control of Mechatronics Lines Served by Complex Autonomous Systems"

_sensors, 2019, doi:10.3390/s19153266_

Round 1

Reviewer 1 Report

The work deals with an extended modeling and control approach of mechatronics lines served by collaborative, complex autonomous systems, able to work as dedicated or shared resources. The outcomes of the approaches are SHPN models associated with A/DML processes served by CASs.

The authors were thorough in the presentation of their work, and the paper is well written, and the results seem correct.  I must say that I enjoyed reading the article.

However, I have some concerns about the work: 

1)    In the abstract, the main idea and contribution must be clarified. 

2)    The authors claim in the abstract that their research aims to create an off-line framework simulation of SHPN models to highlight the evolution of states, the viability of the task planning, and the possible bottlenecks. However, I wonder about the real-time capabilities for the solution. 

3)    The authors claim the existence of other approaches.   Nevertheless, the manuscript contains only results obtained with the proposed method without a performance comparison between the presented work and the previous work. I understand the inherent difficulty in comparing results, due to the need for simulation or experimental data; nevertheless, I suggest the authors offer a more thorough comparison. 

4)    I suggest improving the resolution of the figures (e.g. Figure 2,3,11,15)

Author Response

Changes, suggested by reviewers, were introduced in the original manuscript with Track Changes activated.

1.      In the abstract, the main idea and contribution must be clarified 

The Abstract has been reformulated and clarifies the main aspects addressed in the paper as well as the added value of the paper.

Page 1, lines 12 – 33:

The aim of this paper is to reverse an assembly line, to be able to perform disassembly, using two complex autonomous systems (CASs). The disassembly is functioning only in case of quality default identified in the final product. The CASs are wheeled mobile robots (WMRs) equipped with robotic manipulators (RMs), working collaboratively or in parallel. The reversible assembly/disassembly mechatronics line (A/DML) assisted by CASs has a specific typology and is design by specialized hybrid instruments belonging to Petri Nets class, precisely Synchronized Hybrid Petri Nets (SHPN). The need of this type of models is justified by the necessity of collaboration between the A/DML and CASs, both having characteristics and physical constraints that should be considered and compatible. Firstly, the paper proposes the planning and scheduling of tasks necessary in modelling stage as well as in real time control. Secondly, two different approaches are proposed, related to CASs collaboration: a parallel approach with two CASs have simultaneous actions: one is equipped with robotic manipulator, used for manipulation, and the other is used for transporting. This approach is correlated with industrial A/D manufacturing lines where have to transport and handle weights in a wide range of variation. The other is a collaborative approach, A/DML is served by two CASs used for manipulation and transporting, both having simultaneous movements, following their own trajectories One will assist the disassembly in even, while the other in odd workstations. The added value of this second approach consists in the optimization of a complete disassembly cycle. Thirdly, it is proposed in the paper the real time control of mechatronics line served by CASs working in parallel, based on the SHPN model. The novelty of the control procedure consists in the use of the synchronization signals, in absence of the visual servoing systems, for a precise positioning of the CASs serving the reversible mechatronics line.

2.      The authors claim in the abstract that their research aims to create an off-line framework simulation of SHPN models to highlight the evolution of states, the viability of the task planning, and the possible bottlenecks. However, I wonder about the real-time capabilities for the solution

In the paper, lines 138-157 justify the necessity of HPN models for design of real-time control of A/DML served by CASs:

Page 4, lines 138 – 157:

The need for the SHPN model is justified by the necessity of collaboration between the mechatronics line and the CASs that serve it. Precisely in this approach, the Hybrid Petri Net- HPN, obtained from synchronized HPN without synchronized signals from sensors, is design, simulated and tested in autonomous mode. The compatibility is needed because the mechatronics line and CASs have characteristics and physical constraints that should be considered.

The proposed HPN model is indispensable for simulation and represents the preceding stage of real- time control implementation. As a result of simulation of the HPN model, it is possible to monitor the evolution of the integrated system, A/DML served by CASs, in the state space, as a result of the transients. Therefore, the evolution is consistent with the constructive elements. The inputs of HPN, imposed in modelling stage are: the scheduling of the operations on A/DML, the timings of those ones, the distances and the CASs movement timings, the manipulation timings for each operation, the estimated precise positioning times of the manipulator for taking the piece from the disassembly location and storing it in the corresponding warehouse.

The precise positioning times represent a major uncertainty in our approach because of the existing constructive constraints that could compromise the real- time control. The existing solutions for this problem are based on eye to hand or eye in hand servoing systems. The implementation of this type of control represent for us a target in the nearest future. Until then, we propose for real time control implementation, the HPN model improved with synchronization signals, able to trigger the transitions of manipulator for its precise positioning for take-off the piece from the disassembly station and storing it in the warehouse.

3.      The authors claim the existence of other approaches.   Nevertheless, the manuscript contains only results obtained with the proposed method without a performance comparison between the presented work and the previous work. I understand the inherent difficulty in comparing results, due to the need for simulation or experimental data; nevertheless, I suggest the authors offer a more thorough comparison. 

In our previous publications, cite in the paper, were proposed different theoretical approaches to modelling A/DML served by mobile robots equipped with manipulators.

Our current research focuses on the objectives of the on going research project: PN-III-P1-1.2-PCCDI-2017-0290- Intelligent and distributed control of 3 complex autonomous systems integrated into emerging technologies for medical-social personal assistance and servicing of precision flexible manufacturing lines.

In this context, it was design and implemented the control of an assembly/ disassembly mechatronic line served by complex autonomous systems with parallel actions. Thus, it was necessary to update the previous theoretical approaches from the perspective of supervised control implementation.

4.      I suggest improving the resolution of the figures (e.g. Figure 2,3,11,15)

The resolution of the figures indicated has been improved.

Reviewer 2 Report

This article proposes an extended approach to assembly/disassembly mechatronics lines control, with integrated complex autonomous systems (CASs), working collaboratively or in parallel. Timed Petri Nets (TPN) and Synchronized Hybrid Petri Nets 16 (SHPN)  are used for modelling this type of systems. The paper extensively reports on the off-line simulation of SHPN models.

While the paper is generally clearly written and easy to follow, its contribution appears weak and incremental in relation to the works cited by the authors, including their own (e.g., reference [5] is widely cited throughout the paper). More specifically, quite elaborated and precise models of the considered manufacturing systems are developed and presented, but it is not clear what the use of such models is. Why such modelling is necessary? Related to this point, I found the state-of-the-art not informative about the issues in modelling and control of such lines – maybe a more critical point of view would help to better identify the paper’s merit.

The authors present closed-loop simulation results, but they do not mention any control design procedure, which could possibly make use of the proposed modelling. The control strategy remains mainly descriptive, control objectives are only described, but not formalized; therefore, it is almost impossible to assess if they are validated or not by the simulation results. Usually, such results are discussed against the imposed performance.

For example, the authors mention in the conclusion that “These Petri nets were correlated with tasks scheduling and with no delay or waiting times restrictions. The synchronized times obtained for each task are equivalent with control strategy implementation avoiding collisions, bottleneck or inefficient use of CAS resources.”, but the paper does not contain enough information allowing to draw this conclusion.

In conclusion, the paper cannot be published in its present form. I strongly recommend the authors to address the above issues before the paper can be resubmitted.

Author Response

Changes, suggested by reviewers, were introduced in the original manuscript with Track Changes activated.

1.      This article proposes an extended approach to assembly/disassembly mechatronics lines control, with integrated complex autonomous systems (CASs), working collaboratively or in parallel. Timed Petri Nets (TPN) and Synchronized Hybrid Petri Nets 16 (SHPN)  are used for modelling this type of systems. The paper extensively reports on the off-line simulation of SHPN models.

While the paper is generally clearly written and easy to follow, its contribution appears weak and incremental in relation to the works cited by the authors, including their own (e.g., reference [5] is widely cited throughout the paper). More specifically, quite elaborated and precise models of the considered manufacturing systems are developed and presented, but it is not clear what the use of such models is. Why such modelling is necessary?. Related to this point, I found the state-of-the-art not informative about the issues in modelling and control of such lines – maybe a more critical point of view would help to better identify the paper’s merit.

Lines 138-157 justify the necessity of HPN models for design of real-time control of A/DML served by CASs.

Page 4, lines 138 – 157:

The need for the SHPN model is justified by the necessity of collaboration between the mechatronics line and the CASs that serve it. Precisely in this approach, the Hybrid Petri Net- HPN, obtained from synchronized HPN without synchronized signals from sensors, is design, simulated and tested in autonomous mode. The compatibility is needed because the mechatronics line and CASs have characteristics and physical constraints that should be considered.

The proposed HPN model is indispensable for simulation and represents the preceding stage of real- time control implementation. As a result of simulation of the HPN model, it is possible to monitor the evolution of the integrated system, A/DML served by CASs, in the state space, as a result of the transients. Therefore, the evolution is consistent with the constructive elements. The inputs of HPN, imposed in modelling stage are: the scheduling of the operations on A/DML, the timings of those ones, the distances and the CASs movement timings, the manipulation timings for each operation, the estimated precise positioning times of the manipulator for taking the piece from the disassembly location and storing it in the corresponding warehouse.

The precise positioning times represent a major uncertainty in our approach because of the existing constructive constraints that could compromise the real- time control. The existing solutions for this problem are based on eye to hand or eye in hand servoing systems. The implementation of this type of control represent for us a target in the nearest future. Until then, we propose for real time control implementation, the HPN model improved with synchronization signals, able to trigger the transitions of manipulator for its precise positioning for take-off the piece from the disassembly station and storing it in the warehouse.

2.      The authors present closed-loop simulation results, but they do not mention any control design procedure, which could possibly make use of the proposed modelling. The control strategy remains mainly descriptive, control objectives are only described, but not formalized; therefore, it is almost impossible to assess if they are validated or not by the simulation results. Usually, such results are discussed against the imposed performance.

The reviewer’s observations are justified. Consequently, it was introduced, in the actual form of the paper, Real-time control of CASs based on kinematic model section (lines 395 – 427). This presents the kinematic model used for CASs control, precisely trajectory-tracking sliding mode control.  Also, additional clarification was introduced in 5.2. Control structure in LabVIEW and graphic user interface (GUI) section (lines 428 – 440)

Page 18, lines 395 – 427:

5.1. Real-time control of CASs based on kinematic model

For controlling CASs, trajectory-tracking, sliding mode control (TT-SMC) is, presented in [25]. The CASs: 2DW/1FW, Pioneer 3-DX and 2DW/2FW, Patrol Boot, presented in Figure 15, are controlled to follow a desired trajectory with an imposed velocity.

:

.

Page 20, lines 428 – 440:

5.2. Control structure in LabVIEW and graphic user interface (GUI)

The framework described in previous sections and the offline simulation results allow us to apply a control strategy. The A/DML reversible line served by two parallel CASs makes possible the operation, synchronization and real-time control of flexible manufacturing process, for a given production batch.

Assembly tasks synchronization, testing, decision support and disassembly in case of default issue, are controlled in real time mode, using LabVIEW software. The application design with this tool receives the monitored signals by sensors mounted along mechatronics line, trough data acquisition board (DAQ) and programmable logic controller (PLC) These signals are then used to start or stop the execution of certain tasks according to the planning and optimization goal. The hybrid modelling and model tests were described in the sections above, were need to correlate the dynamic discrete evolution of mechatronics line with continuous evolution of mobile platforms. These approach represents, among other, an added value point of this paper.

3.      For example, the authors mention in the conclusion that “These Petri nets were correlated with tasks scheduling and with no delay or waiting times restrictions. The synchronized times obtained for each task are equivalent with control strategy implementation avoiding collisions, bottleneck or inefficient use of CAS resources.”, but the paper does not contain enough information allowing to draw this conclusion.

Conclusion section of the paper, in the actual form, was reformulated to be in line with the modifications made.

Page 25, lines (lines 518 – 537):

The added value of the paper is to reverse an assembly line, to be able to perform disassembly, using two CASs. In this goal, the characteristics and physical constraints of A/DML and of the two CASs serving it have been put into relation with the constraints of disassembly, transport and storage processes. In the same time, the physical characteristics of CASs have been correlated with the physical characteristics of the manipulated piece. These aspects lead the researchers towards two approaches describing the interactive working of CASs: parallel and collaborative modes.

In modelling approach, the interactive “parallel” actions of the CASs were introduced as solution for disassembly processes of heavy and large manipulated parts, while the collaborative interactive actions of the CASs was proposed to optimize the time for a complete disassembly cycle. The HPN modelling of A/DML served by two CASs has been defined based on the tasks planning and scheduling, proposed in the paper. The SHPN models obtained have demonstrated the need of tasks CASs’s synchronization with the sequential tasks on A/DML. This type of control strategy is related to industrial processes assisted by CASs, in absence of precise positioning visual servoing systems.

The control strategy was implemented for A/DML served by CASs with parallel actions. The supervised control proposed in the paper, have been synchronized the A/DML mechatronics line with CASs’s manipulations and CASs’s movements in SM-TT control. It was design in LabVIEW and the results of the real- time control tests have been presented in the paper. The work in progress research directions are oriented towards the implementation of real- time control of A/DML served by CASs with collaborative actions, in a hybrid and hierarchical control architecture.

Round 2

Reviewer 2 Report

Authors have addressed all the concerns in the revised form of their article.